# Mental Health Nurses’ Tacit Knowledge of Strategies for Improving Medication Adherence for Schizophrenia: A Qualitative Study

**DOI:** 10.3390/healthcare10030492

**Published:** 2022-03-07

**Authors:** Yao-Yu Lin, Wen-Jiuan Yen, Wen-Li Hou, Wei-Chou Liao, Mei-Ling Lin

**Affiliations:** 1Department of Nursing, Tsaotun Psychiatric Center, Ministry of Health and Welfare, Nantou 542, Taiwan; yylin@ttpc.mohw.gov.tw; 2College of Nursing, Chung Shan Medical University Hospital, Taichung 402, Taiwan; wyen@csmu.edu.tw; 3College of Nursing, Kaohsiung Medical University, Kaohsiung 807, Taiwan; wlhou422@gmail.com; 4Department of Medical Research, Kaohsiung Medical University Hospital, Kaohsiung 807, Taiwan; 5Internship Counselling Office, Taichung School for the Visually Impaired, Taichung 421, Taiwan; jou@cmsb.tc.edu.tw; 6Department of Nursing, HungKuang University, Taichung 433, Taiwan

**Keywords:** medication adherence, mental health nurses, tacit knowledge, schizophrenia

## Abstract

Non-adherence to medication among patients with schizophrenia is an important clinical issue with very complex reasons. Since medication administration is an essential nursing responsibility, improving strategies for patient medication compliance must be fully understood. This study aimed to explore the strategies mental health nurses (MHNs) implement in clinically improving patients with schizophrenia and to describe the nurses’ tacit knowledge of application strategies. A qualitative study with purposeful sampling was used. Twenty-five experienced MHNs in a psychiatric hospital in central Taiwan were given an in-depth interview. The texts were content-analyzed using NVivo 12 Pro software. MHNs promote medication adherence among patients with schizophrenia using the following strategies: establishing a conversational relationship, overall assessment of non-adherence to medication, understanding the disease and adjusting the concept of medication, incorporating interpersonal connection feedback, and building supportive resources. This study explored the strategies of MHNs that incorporated knowledge in managing treatment adherence in patients with schizophrenia. The findings add knowledge to clinical nursing practice about medication adherence among patients with schizophrenia.

## 1. Introduction

Adherence to medication is essential but challenging in the psychiatric profession [1,2]. A meta-analysis including 35 studies that reported a pooled estimate of medication non-adherence found that the non-adherence rate in schizophrenia, major depression, and bipolar disorder was 56%, 50%, and 44%, respectively [3]. Antipsychotic medications are effective in successfully preventing relapses if taken regularly [4,5]. Schizophrenia is a lifelong disease with the lowest adherence to medication.

Non-adherence to medication in schizophrenia is a multifaceted issue. Previous studies have found that this non-adherence is associated with first onset, young age, lack of insight, negative attitude to medication, side effects of the medication, social support, medication alliance, alcohol, and substance abuse, among others [6,7]. In addition, a poorly planned discharge, post-discharge environment, and poor therapeutic alliances, among others, can also contribute to non-adherence [8,9,10].

Antipsychotic adverse effects may also occur more in compliant patients because patients with poor adherence are not regularly taking their medications and do not, therefore, experience side effects [5,11]. A study interviewed pharmacists, psychiatrists, and nurses to explore how they help patients manage their medication and found that the topmost concerns were understanding patients’ beliefs about medication and systematically monitoring side effects [12]. The findings indicate the need for a regular follow-up evaluation to simplify medication treatment and reduce the problem of patients taking complex medications to increase adherence [13].

Building a good relationship with the patient is very important in improving medication adherence. A good therapeutic alliance or therapeutic interpersonal relationship between mental health professionals and patients is positively related to improved adherence [5,14,15]. The connotation of a therapeutic alliance includes cooperation, an emotional connection between the therapist and the patient, and common goal setting [16]. Building a trusting relationship with the patient is also the most commonly used strategy used by nurses, which involves listening to and interpreting their needs and concerns [17]. However, many patients with schizophrenia are not hospitalized voluntarily, and their relationship with medical professionals is not spontaneously established, which makes implementing measures to enhance adherence with medications difficult for nurses. Furthermore, no relevant studies have been conducted.

In order to improve medication adherence among patients with schizophrenia, MHNs (mental health nurses) provide psychological education to increase patients’ understanding of the characteristics of the disease and recognition of medications [18], cognitive behavior therapy [19], and adopted motivation interviewing to increase motivation to take medications [20] in the clinical setting. Otherwise, long-acting antipsychotic injections can be administered to assist oral medication to decrease relapse [21]. Healthcare professionals encourage patient involvement in shared decision making with their physician to express their preference and opinion in treatment selection to increase their adherence to medication [22]. Inviting family caregivers involved in the treatment of patients with schizophrenia is one strategy to increase adherence to medication [9]. Some studies suggest that social support is also related to medication adherence [5,9,11], but some studies report that it is unrelated [14]. Nurses have frequently conducted psychoeducation for families in practice [23], which means that social networks can support patients and encourage them to adhere to their treatment. Involving family for social support may vary with culture. Therefore, the patient’s condition or preference during the intervention should be considered before involving the family.

Nurses have a positive influence and can help patients change their attitudes toward disease and enhance their insights to increase medication adherence [24]. Previous studies aimed to clarify the factors that influence medication adherence in psychiatric patients. However, there are two problematic aspects in clinical practice. First, the quantitative study separates the variables into a part of medication adherence that cannot be understood in the whole view context of increasing medication adherence. Second, previous studies rarely demonstrated the “how” aspect; specifically, how the influences on medication adherence embody nurses’ experiences. The 3D creativity management theory considers the within-discipline expertise, out-of-discipline knowledge, and a disciplined creative process to explain how creativity and innovation are manifested throughout information processes [25,26]. Nurses have the longest and most frequent contact with patients. Moreover, patients’ adherence to medicine must consider the overall context in clinical practice. The experiences of MHNs may embody the tacit knowledge gained in actual practice; however, these experiences have not been highlighted and consolidated in previous studies.

### Aim

This study has two goals. First, it sought to explore the strategies employed by MHNs to improve medication adherence in clinical practice. Second, it aimed to describe MHNs’ knowledge of facilitation strategies in medication.

## 2. Methods

This study was based on qualitative exploratory methodology and used in-depth interviews to collect information on nursing experiences and strategies for medication adherence in mental illness. Ethical approval was obtained from the research ethics committee (TTPC 108026) and the administrative committee of a psychiatric hospital in Taiwan. All participants provided signed informed consent on their understanding of the study and to the audio recording of their interviews.

### 2.1. Sample

All participants were selected from a psychiatric hospital in central Taiwan. The corresponding author conducted all interviews. The participants were accompanied by MHNs who were over the age of 20 years and had at least 1 year of nursing experience. One-on-one interviews were conducted for data collection. The interview began with questions on the participant’s recognition and experience of non-adherence, followed by questions that elucidated how they dealt with those situations.

### 2.2. Data Collection

Purposeful sampling was conducted on 25 MHNs in this study. The researcher kept the interview questions in mind and used them flexibly to avoid interfering with the interview process. The interviews were conducted from February to July 2020. An in-depth interview included questions focusing on the strategies employed by MHNs to improve medication adherence and describe the knowledge of facilitation strategies in medication. These interviews used an interview guide with questions that were developed a priori by the authors based on their review of the literature and clinical experience: “Could you describe care experience about non-adherence medication of schizophrenia?”, “Based on your clinical experience, could you describe the reasons for the non-adherence medication for schizophrenia?”, and “Could you give me some examples of how to increase non-adherence medication for schizophrenia?”. Additionally, the following questions were asked about the context, in order to elicit tacit knowledge of medication adherence in nurses’ experiences: “How does this happen?” or “What causes this to happen?”.

Each participant was interviewed one to two times. The interviews lasted for 45–65 min. All interviews were audio-recorded and transcribed verbatim. Possible identifying information was deleted from the data, and codes were used to replace the names of participants. Data were kept anonymous, and coding numbers were used during analysis to ensure they could not be linked to any personal information.

### 2.3. Data Analysis

A 7-step inductive qualitative content analysis was performed [27]. Data analysis began after the first interview and simultaneously continued with subsequent interviews until data saturation was reached for the purpose of this research and no new relevant information could be obtained [28,29]. The data were saved in Word files before being uploaded to NVivo 12 Pro software (QSR International, Melbourne Australia) for analysis.

Each unit of analysis was given meaning units from the text. Descriptions of participants who used strategies to increase medication adherence for mental illness were as detailed as possible. Preliminary labels were given to the meaning units and were grouped based on similarities and differences. The codes from the data were divided into subthemes, and these themes were labeled according to the content. The subthemes were further compared and grouped based on similarities and differences. The themes were named based on content characteristics. The researchers continued to analyze the data until data saturation had been reached.

### 2.4. Rigor and Credibility

To ensure rigor [30], purposive sampling was conducted, and the participants enrolled all had experience of promoting medication adherence in psychiatric mental illness. The researcher (M.L.L.) had a background in phenomenology and experience in qualitative research in the field of mental health nursing. The researcher established a relationship of respect and trust with each participant and maintained intersubjectivity. All participants provided rich, diverse perspectives of the phenomena in this study. To enhance transferability, interview questions were used to assist the researcher to encourage the participants to recall their caring experience of medication non-adherence with detailed descriptions to obtain extensive data. Regular research meetings among researchers were held to examine the verbatim transcriptions to ensure the quality of the interviews. The text was preliminarily coded and categorized independently and manually (MLL). The codes, subthemes, and themes were checked and discussed (WCL). If the subthemes or themes varied, the authors modified the list and read the transcripts to confirm the participants’ views until an agreement of categorization among researchers was achieved. The participants provided their comments after reviewing the summary of the theme, sub-theme, and summary of the final results. Regular meetings with the authors were held. To ensure confirmability, peer debriefings among researchers were adopted to discuss the findings, emerging themes, and interpretations of data. Reflective diaries were kept to assist the researcher to maintain neutrality during the data analysis process.

## 3. Results

The demographic characteristics of the 25 MHNs are presented in Table 1. The following were identified as the strategies MHNs used to promote medication among patients with schizophrenia: establishing a conversational relationship, overall assessment of non-adherence to medication, understanding the disease and adjusting the concept of medication, incorporating interpersonal connection feedback, and building supportive resources (Table 2).

### 3.1. Establishing a Conversational Relationship

Establishing a conversational relationship means that nurses create afeeling of safety and trust for patients and establish a basis for dialog with patients about taking medication. Subthemes of this topic include concerning daily life, expressing care nonverbally, lenience and being-with, and handling patients’ characters.

#### 3.1.1. Concerning Daily Life

Nurses interact with patients in their daily lives to reduce the sensitivity of patients and to create a starting point for interactions. Only by gaining trust can MHNs further discuss the core problems of the disease. Participant 17 thinks that providing care in daily life is not intrusive and avoids the patient’s symptoms. She said:

“If his (patient) symptoms are really bad, or he has delusions, don’t be too deliberate. Continue providing routine care every day. You can ask: How are you (patient) sleeping lately or how are you eating? He may feel that this is just a common concern. The patient may be willing to talk about why he doesn’t (want to) take his medication further.”

MHNs start with normal contact, and only after establishing trust can they talk to the patient about problems with their disease and medication status.

#### 3.1.2. Express Care Nonverbally

Non-verbal communication of care acknowledges that patients are more sensitive to non-verbal expressions of care. MHNs can convey their concern for patients through non-verbal communication, which helps to build relationships. Participant 9 discusses this as follows:

“Actually, language is a secondary form of communication. What’s more important is the nurse’s attitude, expression in the nurse’s eyes, tone of voice, and spending time with them. That can help the patient feel safe to talk about their illness.”

Given the importance of non-verbal communication in relationship building, MHNs thus try to communicate nonverbally.

#### 3.1.3. Lenience and Being-With

Containment and being-with mean that nurses respond to the patient’s negative emotions through listening and empathy to convey understanding and acceptance to the patient. Participant 9 recalled facing psychiatric patients complaining about medication problems:

“When he (patient) may have some complaints, I just comfort him. He was complaining about the discomfort of taking medicine. First, I listened to his complaints. After he complained, he talked about taking medicine slowly.”

Participant 10 thought that, given enough time, the patient will gradually vent out their emotions, which will help to establish the relationship. He said:

“You should give the patient plenty of time to talk nonsense or complain and vent the accumulated emotions and pressure. He (the patient), in this way, will more likely develop trust in you.”

Nurses can show empathy, listen, and accept behaviors to deal with patient’s negative emotions and earn the patient’s trust in order to talk about medications.

#### 3.1.4. Handling Patients’ Characters

Handling patients’ characters means nurses should appropriately adjust the interaction according to the different personalities and situations with the patients to maintain a friendly interaction. At the beginning of the interaction between Participant 11 and a patient, an adjustment of the interaction according to the patient’s characteristics facilitated the patient’s cooperation in therapeutic activities after the relationship was established. She said:

“There are some patients who are more perceptive. It is better to give him a little flexibility rather than confront him. Then, he will be more willing to accept and cooperate in the follow-up. We learn how to use the patient’s traits to take care of him.”

Nurses should have sensitivity during the interaction. Participant 17 thought that they should not talk about medication too directly. She said:

“At the beginning, you can’t directly talk about medicine. No! Otherwise, you will not be received well. The patient may instead say that they don’t use it anymore. Patients will disagree with you. You will be rejected.”

The nurse is unable to carry out the follow-up treatment plan if it triggers negative emotions in this case.

### 3.2. Overall Assessment of Non-Adherence to Medication

Non-adherence to medication involves various factors, and the nurse should assess the medication status and difficulties. The subthemes of this include evaluating beliefs about taking medication from the context, tactfully confirming side effects, and assessing the ability of the family to support medication.

#### 3.2.1. Evaluating Beliefs about Taking Medication from the Context

Evaluating the beliefs about taking medication from the context means that the nurses evaluate the reasons why the patient is disinclined to take medication to use as a reference for intervention. Participant 21 described a clear reason:

“The patient’s lack of desire to take medicine must be based on reasons. If the patient can specify his reasons, we can assist him in eliminating them to improve his motivation to take medicines.”

The nurse should also confirm the medication status from the patient’s relatives and friends. In this regard, Participant 12 said:

“I listened to the patients and the family members to see if there are any points of fit because they actually live together.”

In addition to assessing medication, the nurse should also confirm from their relatives observervations whether patients take medication. The assessment status is used as the basis for interventional medication by understanding the context.

#### 3.2.2. Tactfully Confirm Side Effects

Nurses know that side effects are one of the reasons for non-adherence to medication. They take indirect or tactful methods to clarify the side effects experienced by patients. Participant 6 described an example using postural hypotension:

“If the patient is taking medicine that causes postural hypotension, we will ask: ‘Do you usually have it?’ or ‘In addition to the uncomfortable things you said, do you still feel dizzy?’ We will not directly say that this is a side effect of the medicine.”

Participant 19 also mentioned:

“We will avoid saying we notice the patient’s worry, such as tremor of hands. I will not ask him directly, ‘Do your hands tremor?’ Instead, I will ask, ‘Do you feel uncomfortable after taking the medicine?”

MHNs use euphemistic methods to clarify and evaluate to prevent patients from hesitating or ceasing to take medicine due to side effects.

#### 3.2.3. Assessing Ability to Support Medication from Family

To determine the knowledge of family members in supporting medication adherence, the family’s ability and individual problems of the family members as medication supervisors are assessed to confirm the patient’s medication problems. Participant 16 described the assessment of the family members’ ability and concern about medication adherence:

“Sometimes family members have to work and can only prepare medicines for him to take. Sometimes family members are afraid of the patient’s anger, fearing that the patient will be upset by being urged to take the medicine. Nurses need to evaluate the abilities and attitudes of family in supervising the patient in adhering to medication.”

The family members are the main caregivers of patients with mental illness, and nurses need to evaluate the ability of a family as a resource in medication adherence.

### 3.3. Understanding Disease and Adjusting the Concept of Medication

Nurses use different viewpoints to help patients understand disease, explain the pros and cons of medication, and improve patients’ willingness to take medication. Subthemes include an analogy to an acceptable disease, the pros and cons of taking medication, identifying and managing medications, and reducing influence from side effects.

#### 3.3.1. Analogy to an Acceptable Disease

Creating an analogy to an acceptable disease means that the nurse compares the treatment of psychiatric medication to a generalized disease that can be accepted to increase the patient’s willingness to take medicines. Participant 3 explained that taking psychiatric drugs is similar to taking medication for a general physical illness to increase motivation in taking medication:

“Some patients have no insight. I use analogy to physical diseases. I will tell patients: ‘Taking this medicine (antipsychotic) is just like taking medications if you have high blood pressure or heart disease.’ I also tell the patient, ‘Even us nurses have to take medicines when we are sick.”

Participant 12 used acceptable generalized symptoms as a starting point for the explanation:

“I think that discussing symptoms of the disease that are acceptable to the patient, such as insomnia, and asking them to confirm with continued treatment whether the symptoms improve can persuade him to take the medicine.”

MHNs can help improve the patient’s acceptance of symptoms and medications by comparing the treatment process to those of general illnesses.

#### 3.3.2. The Pros and Cons of Taking Medication

The nurse discusses the pros and cons of taking medication to emphasize that failure to do so will cause the return of symptoms and relapse. The nurse reminds the client of the consequences and losses of not taking medication. Participant 9 positively reminded a patient of the changes after the medication:

“Let the patient compare the results of adherence and non-adherence, and ask them to think about what is good for them.”

To encourage continuous medication treatment, Participants 21 said:

“Let the patient know that with treatment, their situation would be more stable. Without treatment, they may be sick or hospitalized all the time, unable to work, unable to maintain their marriage, and the family will eventually be affected.”

The nurse can enumerate the benefits of taking the medicine to avoid losing one’s well-being due to the influence of symptoms.

#### 3.3.3. Identifying and Managing Medications

Identifying and managing medications refers to improving motivation to take medications by understanding the benefits of drugs for disease control. The improvement in symptoms after patients administer the medicine is used as the premise to increase their awareness regarding medications. Participant 20 stated:

“I will inform patients about the effects and side effects of medications. The main purpose is to let the patients know how medicine can help stabilize the disease.”

MHNs can also assist in the administration of medicines so that patients can take medicines conveniently. Participant 12 said:

“There are some patients who will still be unable to identify their medicines clearly. I can help them concentrate the medicine and tell them: ‘this package is your after-breakfast medication.”

MHNs can assist patients in organizing medications to improve medication adherence behaviors and convenience.

#### 3.3.4. Reducing Influence from Side Effects

Side effects are one of the reasons that cause patients to stop adhering. One of the strategies used by nurses to improve patients’ medication adherence is to help reduce the side effects via providing information or assisting the physicians to adjust the medications that affect patients’ confidence in taking the medication. Participant 5 provided suggestions to alleviate side effects after speaking with the patient:

“I will provide nursing knowledge and inform patients that if side effects are present, the medication can be adjusted to relieve discomfort.”

MHNs can act as agents for the patient. Nurses can communicate with the physician the need to adjust medication and reduce the impact of side effects. Participant 2 said:

“Some patients need to work, and the drugs make them groggy and sleepy. I can ask the doctor to make some adjustments in the medication regimen.”

MHNs can provide information on reducing side effects and also serve as a communication bridge between patients and physicians to increase patient willingness to take medication while helping decrease the influence of side effects.

### 3.4. Incorporate Interpersonal Connection Feedback

Incorporating interpersonal connection feedback means helping patients perceive the benefits and responsibilities of taking medications through feedback of the wellness associated with taking medications as prescribed. Subthemes include positive affirmations for taking medication, describing illness anecdotes from peers, and consideration of the burden on their loved ones.

#### 3.4.1. Positive Affirmations for Taking Medication

Positive affirmations for taking medication means providing encouragement to the patient to boost the patient’s willingness to continue taking medication. Participant 4 provided patients with feedback on taking medication through affirmation cards:

“The patient likes the star. I made cards and wrote blessings on the back of the cards. After the patient takes the medicine, I give them the card as positive affirmation for adhering to medication.”

Positive feedback improves the motivation of patients to adhere to medication, thereby enhancing medication compliance.

#### 3.4.2. Describing Illness Anecdotes from Peers

The nurse invites the patient’s peers to describe their experiences of the benefits of taking the medicine to enhance the patient’s motivation to do the same. Participant 12 described her experience:

“I invited patients with the same disease to talk about their experiences of taking the medicine. The invitees told the hesitant patient, ‘You must take it! I can sleep better and I don’t have a bee in my bonnet.’ Hearing anecdotes from peers has a significant effect on persuading patients to do the same.”

Hearing the experiences of peers with the same can increase motivation for adherence to medication.

#### 3.4.3. Consideration of the Burden on Their Loved Ones

Consideration of the burden of the disease on their loved ones entails the nurse reminding the patient to pay attention to the effort of their caregivers in helping them manage their illness. Participant 7 recalled:

“Tell the patient that if they take their medicine, they will be less likely to be hospitalized and that they can also make their mother feel more at ease.”

Participant 16 further reminded patients to value the hard work of their family members in helping them manage their illnesses. Only stabilizing their symptoms can reduce the burden on their family. She said:

“I tell the patient: ‘Your mother is worried about your health, but she is also getting older. If you are willing to take the medicine and to manage your symptoms, then you will be also helping unburden you mother.”

MHNs can remind patients that their illness not only affects themselves, but also increases the burden on their family. Taking medications as directed helps to stabilize symptoms and reduce caregiver burden.

### 3.5. Building Supportive Resources

The nurse evaluates how to help the patient comply with the medication regimen by changing the patient’s participation in medical care and/or oral medication status or using family and community resources. The subthemes of this intervention include encouraging shared decision making, using long-acting antipsychotics, inviting family to monitor symptoms and promote medication adherence, and referral to related resources.

#### 3.5.1. Encourage Shared Decision-Making

MHNs encourage patients to communicate with their physicians in order to talk about medicine and fight for their best interests. Participant 19 said:

“You can adjust the medication during the hospitalization, and you can talk to the attending doctor. You can also communicate with your doctor and make adjustments during outpatient visits.”

Nurses can encourage patients to express their thoughts on the medications prescribed to them and to get involved in consultation with doctors to find the most suitable medical treatments for themselves.

#### 3.5.2. Using Long-Acting Antipsychotics

The use of long-acting antipsychotic injections can help stabilize the illness and its symptoms for a longer duration. Participant 1 mentioned that changing the drug dosage form is one of the strategies to improve medication:

“I will tell the patient during the hospitalization: ‘Long-acting antipsychotics can be injected once a month. If you forget to take the oral medicine, you will not relapse too quickly which can cause you to be hospitalized.’ Oral medications are more prone to non-adherence. Long-acting injections help remedy this and/or increase compliance.”

#### 3.5.3. Inviting Family to Monitor Symptoms and Promote Medication Adherence

Families can help to monitor the patient’s symptoms and medication adherence at home. Therefore, improving family members’ knowledge of symptoms and drugs is helpful. The nurse can remind and assist family members in this. Participant 16 let the family members know about the symptoms and medication status:

“I will tell the family to pay attention to the signs of the patient’s illness, for example, to see if the patient is starting to suffer from insomnia or emotional instability, and to pay attention to his medication status.”

Family members confirm that medication and illness status contribute to adjuvant treatment. Participant 22 said:

“I will mention to the family that you have to observe the patient’s medication status and confirm the remaining amount of medication to prevent any disruptions.”

Family members have an important supporting role in enhancing medication compliance. Nurses can teach family members to identify the signs of onset and confirm medication.

#### 3.5.4. Referral to Related Resources

A pre-hospital search for related resources can help stabilize the patient’s condition and improve compliance with medications. Participant 2 mentioned that when assessing patients for poor drug compliance, follow-up resources should be considered to continue care. She said:

“If the patient does not cooperate with the medication, he may prepare for referral to a home care, daycare, or another sheltered environment and find a place that can help them supervise medication adherence before leaving the hospital.”

From the perspective of resource linking, patients can continue to take medication after discharge from the hospital.

## 4. Discussion

This study explored the strategies of MHNs that incorporated tacit knowledge in managing treatment adherence in patients with schizophrenia in clinical work. MHNs choose medication adherence strategies based on a judgement that considers patient attitude, side effects, and related resources. The first conditions need to be fulfilled in building a trusting relationship between nurses and patients. There is a complex process based on the patient’s response in the next step. Nurses consider “act” to be a viable and preferable option in terms of the patient’s response that is similar to 3D creativity management theory to explain how creativity and innovations from nurses are involved in promoting adherence to medication in patients with schizophrenia [25]. Nurses nonverbally express sincerity and safety, provide psychoeducation, and go on to build a trustworthy relationship with patients. The core tenet of medication adherence strategies is to avoid ruining relationships with patients. The results indicate that nurses will often select strategies from patients’ feedback, which is a dynamic process, and consider the best benefits for patients. Nurses euphemistically and thoughtfully adjust their behavior to the attitude of the patient, make sure the information is available, evaluate information, and provide related resources to increase medication adherence.

### 4.1. Trust Building

A high level of trust helps facilitate information reception and acceptance. A previous study promoted medication adherence and acknowledged that patients’ beliefs about medications play an important role [12]. The authors agree with this opinion. MHNs, as the first-line psychiatric professionals in the management of mental illness, are well aware that the relationship with the patient can constitute the basis of drug beliefs. It is critical for psychiatric patients to take medications as prescribed. Research has shown that establishing therapeutic alliances with patients can increase the benefit of medications [16]. However, in the early stages of the treatment relationship, few patients are voluntarily hospitalized, and the positions of nurses and patients tend to differ. The key to the role of nurses in enhancing compliance with medications depends on whether the patient can trust the nurse as a starting point for cooperation that maintains dialog and interaction with patients.

MHNs’ “tacit knowledge” is different from “explicit knowledge” of improving mediation adherence in patients with schizophrenia. Explicit knowledge can be easily expressed, written, and transferred from one person to another. In this study, nurses accumulate tacit knowledge that is obtained from the experience and intuition gained during interpersonal interaction based on hands-on experience and in-depth practice analysis, observation, etc. The term “tacit knowledge” was coined by Michael Polanyi [31] to refer to the knowledge that cannot be transferred through formalization (writing or verbalization). Similar to previous research, MHNs also think that good interpersonal relationships between mental health professionals and patients can improve adherence [5,14,15]. Moreover, MHNs strive for cooperation and alliance with patients and avoid triggering negative emotions in the patient that may cause the patient to distance themselves from or reject the nurse in clinical practice. It could be based on the concept of the disease, and the cognition of medications is different between the patient and the nurse at the beginning of the contact. In this regard, the MHNs used a careful probing attitude to confirm the problem of the patient’s adherence to the prescribed medication. Establishing an interpersonal strategy for maintaining conversational relationships to improve compliance with medications cannot be applied as a whole. Given the wide variety of patient personalities, nurses need to flexibly use their experience to slowly grasp the essentials in an interaction.

### 4.2. Information Availability

Nurses help patients identify what information is needed and to deal with side effects to increase medication adherence. MHNs know that patients encounter side effects and this may cause them to stop taking the medication [5,11]. The nurses use tactful methods to assess the patient’s feelings about the side effects in the identification process. They express a caring attitude to ask generalizing questions for the assessment of side effects and symptoms, but they avoid directly mentioning the side effect to the patients. Nurses are concerned about highlighting the side effects that may cause patients to stop taking the medication. Nurses try to lead patients to receive medication and improve medication compliance, but nurses using their knowledge as power should maintain a high degree of self-awareness and caution while using this “invisible” power in professional roles [32]. The MHNs assess side effects in indirect ways to avoid worsening the patient’s apprehension and reluctance to take medication. However, MHNs may neglect to inform patients about side effects.

### 4.3. Information Evaluation

Nurses understand that the patients’ perceptions of medication depend on what subjective values they attach to that concept. Generalization of the disease and making analogies can help the patient to understand the purpose of adhering to medication and help soften or change the patient’s perceptions of antipsychotic medications due to the social stigma of mental illness [33,34]. MHNs can provide positive affirmations and invite family members to support medication adherence, as also suggested in previous studies [5,11]. The authors found that nurses can enlist the help of peers with similar illnesses and more insight to share positive experiences of medication adherence and provide anecdotes that are relevant to the patient. Research plans previously proposed to improve medication adherence are similar [35,36]. However, nurses who have accumulated experience and information through interpersonal exchange can persuade patients to willingly take medication as prescribed by doctors.

The nurse reminds the patient that disease stability can reduce the burden on their loved ones to motivate them to take their medication. This has not been mentioned in previous studies. Even good support from relatives and professionals also motivates patients to continue taking medication [37]. It may be because Confucianism is deeply rooted in Taiwanese society and has expectations of the relationships between people in society [38]. Although this approach may increase the patient’s awareness of taking medicine for others, how long can the effect last? What is the benefit to the patient? Further research is still needed to explore these questions.

### 4.4. Resource Management

A patient’s willingness to adhere to medication is based on a complex system of interlocking. The patient can best understand his body and the improvement of symptoms after taking the medicine, as well as his preferences. The nurse encourages the patient to discuss their treatment with the doctor for shared decision making [39]. The patient advocates for their needs, which can positively affect compliance with medication. Medication management can facilitate patient recovery. Nurses will consider relevant resources for promoting patient adherence to medication. There are several resources available to support patients’ adherence to medication, including inviting family members to monitor symptoms and promote medication adherence, discussing with the physician the provision of long-acting antipsychotics as an injection, and the referral to related community systems. MHNs’ cognition of adherence to medication is not the patient’s business, but needs a system for support.

This study expects to gather information relevant to an increase in medication adherence strategies from MHNs, which can serve as a reference for new colleagues and psychiatric nurses to continue education. Additionally, considering the positive impacts of medical research on not only the patients but also the whole society, MHNs’ strategies for medication adherence need to further comprehensively consider the role of the institution in assisting patients to adhere to medication. The value of this study as a scientific enterprise brings integrity and self-correction in social evolution [40]. Furthermore, it is necessary to provide nurses with sufficient education, training, and counseling. It should be ensured that nurses can receive adequate support. The related systems should also be sufficient for supporting patient adherence medication tracking after discharge, thereby preventing patients from constantly swinging through the revolving door of hospitalization.

The advantage of this study is that way in which samples were selected was a fair and adequate reflection of the study purpose, which maximizes the potential transferability of the study. More qualitative research needs to be undertaken in various contexts and among patients with similar economic statuses to widen the current understanding of the factors that influence medication adherence. While the analysis attempted to remain as close to the interview data as possible, the selection of extracts and analyses inevitably involved subjective interpretation; thus, other interpretations may also exist concurrently.

## 5. Conclusions

This study has responded to how nurses increase medication adherence in patients with schizophrenia from the internalized experiences and practice of nurses through practical exercises. Because there are diverse reasons for nonadherence in patients with schizophrenia, nurses cannot use the same methods in the increased medication adherence process, and need to consider patients’ situations in order to adopt fitting strategies. The establishment of a conversational relationship is the starting point in the overall assessment of the causes of non-compliance with medications, the adjustment of attitudes toward disease and medication, and the promotion of participation with medication. This method of connecting with patients with schizophrenia allows them to better understand their symptoms and achieve medication compliance, which in turn will help stabilize their symptoms and reduce the burden on loved ones. Side effects are an important factor affecting patients’ willingness to take medications. MHNs can actively intervene in reducing side effects and thus reduce non-compliance with medications.

## 6. Relevance for Clinical Practice

This study discusses the strategies used by experienced nurses to improve medication compliance in patients with schizophrenia. The use of seminars and problem-based learning contributed to the rich experience of nurses in medication adherence. To enable nurses to exchange experience, they should be encouraged to continue learning and use their accumulated knowledge when interacting with patients to complement the static clinical guidelines.

## Figures and Tables

**Table 1 healthcare-10-00492-t001:** Sociodemographic characteristics of the participants.

Demographics			*n* = 25
Sex	Female (%) Male (%)	23 (92%) 2 (8%)	
Age (years)		Mean (SD)	39.6 years old (SD 6.1)
Education		College/university	11 (44%)
		postgraduate	14 (56%)
Marital status		Married	19 (76%)
		Single	5 (20%)
		Divorced	1 (4%)
Psychiatric work experience		Mean (SD)	13.4 years (SD 5.9)
Current psychiatric work department		Acute ward	15 (60%)
		Chronic ward	5 (20%)
		Community	5 (20%)

**Table 2 healthcare-10-00492-t002:** Summary strategies of MHNs to promote medication adherence in schizophrenia.

Themes	Subthemes
1. Establishing a conversational relationship	1.1. Concerning daily life 1.2. Express care nonverbally 1.3. Lenience and being-with 1.4. Handling patients’ characters
2. Overall assessment of non-adherence to medication	2.1. Evaluating beliefs about taking medication from the context 2.2. Tactfully confirm side effects 2.3. Assessing ability to support medication from family
3. Understanding the disease and adjusting the concept of medication	3.1. Analogy to an acceptable disease 3.2. The pros and cons of taking medication 3.3. Identifying and managing medications 3.4. Reducing influence from side effects
4. Incorporate interpersonal connection feedback	4.1. Positive affirmations for taking medication 4.2. Describing illness anecdotes from peers 4.3. Consideration of the burden on their loved ones
5. Building supportive resources	5.1. Encourage shared decision making 5.2. Using long-acting antipsychotics 5.3. Inviting family to monitor symptoms and promote medication adherence 5.4. Referral to related resources

## Data Availability

All data were generated at the Tsao Psychiatric Center, Taiwan. The derived data supporting the findings of this study are available from the corresponding author Lin, M.-L. on request.

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
