# Peer review of "Mental Health Nurses’ Tacit Knowledge of Strategies for Improving Medication Adherence for Schizophrenia: A Qualitative Study"

_healthcare, 2022, doi:10.3390/healthcare10030492_

Round 1
Reviewer 1 Report
Adherence to antipsychotic treatment is a current and highly relevant issue due to the numerous complications it entails for patients and the increase in health care costs, with the role of the nurse being very important in patient follow-up. For this reason, the subject of the article is of great importance.
Some things to improve:
- Indicate the abbreviation of MHN (mental health nurses) on line 67, which is where it appears for the first time in the text.
- In the methods section, I miss the variables that were collected in the interviews carried out, as well as whether they are self-elaborated or whether the authors have based themselves on a validated questionnaire.
- Table 1 should be included in the results section and not in the methods section.
- A figure showing the strategies used by experienced nurses to improve medication adherence in patients with schizophrenia is suggested for inclusion.
Author Response
Responses and revisions: Thank you for the suggestions to make our paper clearer.
- We added the expanded form of the abbreviation MHN at its first mention on line 68.
- This study conducted qualitative research that is not based on variables.Therefore, in the Methods section, we added interview questions to make the content clearer.
- Thank you for the correction. We have moved Table 1 to the Results section.
- We have added the Table 2 showing the strategies used by experienced nurses to improve medication adherence in patients with schizophrenia.
Sincere thanks to the Reviewers for their insightful suggestion to reconsider and adjust this research more completely.
Reviewer 2 Report
The reviewer has read this manuscript carefully and assessed the incremental contribution of this paper in terms of the relevance of topic, proper research design, and theoretical/practical implications.
This is a well-written article to explore the tacit knowledge of strategies for improving medication adherence for schizophrenia patients from mental health nurses (MHNs), however; there are a few points that the author(s) may need to provide further clarification:
- As mentioned in the introduction, there have been many studies exploring the issue of non-compliance with medication for patients. I suggest that the author(s) describe in more details about how understanding the experience of MHNs may further contribute to current knowledge.
- The findings of the study are generally similar to those currently known strategies have been used. I suggest the author(s) discuss more about what are the important, new, or uniquely useful strategies found by this study, especially which of them are different to current strategies in improving patient compliance with medication.
I hope my comments helpful. Good luck!
Author Response
Responses and revisions:
Thank you for the suggestions to make our paper clearer.
- Nurses play a crucial role in the medication adherence of psychiatric patients. We added the content of the nurse’s implementation improvement strategy in the Introduction section, which is indicated in red font.
- Although the results are similar to the currently known strategies, the value of this study demonstrates the “how” aspect in the experiences of nurses who implement medication adherence strategies in schizophrenia. We incorporated the value of this study in the Discussion section.
Sincere thanks to the Reviewers for their insightful suggestions to reconsider and adjust this research more completely.
Reviewer 3 Report
Title: Mental Health Nurses’ Tacit Knowledge of Strategies for Improving Medication Adherence for Schizophrenia: A Qualitative Study
Non-adherence of medication among patients with schizophrenia is an important issue with very complex reasons. How to find an effective strategy to assist patients' adherence to medication will be of great help to MHN's in clinical practice.
Reviewers' Comments and Suggestions for Authors
- In the abstract part: The authors addressed using a questionnaire with in-depth interviews. Since the interview guideline is not presented in the manuscript, the researcher should confirm whether to use the interview guideline or the questionnaire to collect data?
- In the introduction section, it is recommended to describe the importance of tacit knowledge for Mental Health Nurses.
- Participants were selected from a psychiatric hospital in central Taiwan. Please describe whether participants are from the same hospital or from different hospitals.
- Nursing therapists and MHN are the same or different. Please establish consistency or provide clarification in the manuscript.
- Please explain that there are five nurses who are current psychiatric work departments in the community. Is their work character the same as acute and chronic medical wards?
- In 2.4. Rigor and Credibility. The research team made many rigorous statements. However, the first author is the main interviewer, and the qualitative training needs to be described further.
- In 3.1.2. Express Care Nonverbally. Participant 9 described: “Actually, language is a secondary form of communication. What’s more important is the nurse’s attitude, expression in the nurse’s eyes, the tone of voice, and spending time with them. That can help the patient feel safe to talk about their illness.” Did the interviewer attempt to dig deep into nonverbal strategies? If so, does MHN share reliable nonverbal methods or strategies?
- In 3.3.4. Reducing Side Effects. This subtheme needs further clarification. Maybe consider adding provide information on reducing side effects.
The article found that MHNs have fairly diverse strategies to improve medication adherence in schizophrenia patients. However, it is still necessary to clarify the reviewer's comments to fully present the research findings.
Author Response
Responses and revisions:
Thank you for the suggestions to make our paper clearer.
- We have changed the term. Thank you for the reminder.
2. We added the description of tract knowledge in the Introduction section to emphasize its importance among mental health nurses.
- All participants were selected from a psychiatric hospital in central Taiwan.
- Thank you for pointing this out. We have used the term “mental health nurses” in the manuscript for consistency.
- There were five nurses who worked in the community, and their role is to provide home care, which involves the assessment of psychopathology and the development of interventions, including educating patients regarding medications. Their work character is the same as a hospital unit.
- Thanks for the suggestion. We added the experience of the author’s background into rigorous statements.
- This is difficult to express in words, but the core value context of a nurse’s nonverbal strategies is that the nurse makes the patient feel safe.
- We review the content and modify the subtheme into ‘Reducing Influence from Side Effects
Sincere thanks to the Reviewers for their insightful suggestions to reconsider and adjust this research more completely.
Reviewer 4 Report
Review for the manuscript titled:
Mental Health Nurses’ Tacit Knowledge of Strategies for Improving Medication Adherence for Schizophrenia: A Qualitative Study
Journal: Healthcare
Date: Feb 15, 2022
The study records and categorizes nurses’ strategies when dealing with medication non-adherence in patients with schizophrenia. The topic is interesting, and the findings have practical value in the treatment process.
While the study was conducted well overall regarding methodology, structure, and language, in my opinion, the categorization of nurses’ experiences and strategies still lacks a conceptual framework to make it more systemic. My rationale and suggestion are as follows.
When examining the interactions between nurses and patients, it should be viewed as a process of information exchange. In the scope of this study, we focus on the patient side (for medication adherence purposes) with several aspects: their trust toward nurses, what kinds of information they need, and how they understand that information. This is particularly important in the case of schizophrenia patients, where information tends to be processed differently compared to normal people (which is also why nurses’ context-based tacit knowledge is crucial).
Thus, I recommend incorporation of the information processing approach; the authors can read about such approach in this article on suicidal ideation and help-seeking behavior: https://www.mdpi.com/1660-4601/18/7/3681
I think it is unnecessary to change the current structure of the paper. Rather, having a new sub-section about information processing in the discussion section would be a nice addition. Detailed suggestions are as follows; I hope the authors find them helpful.
- Trust building: high level of trust helps facilitate information reception and acceptance (here, the information sources are nurses). This is section 1 “Establishing a Conversational Relationship”.
- Information availability: ensuring patients’ access to useful information about medication. Section 2 “Overall Assessment of Non-Adherence Medication” is about identifying what information is needed; Section 3.3 “Understanding Disease and Adjusting the Concept of Medication” is about the information’s content.
- Information evaluation: patients’ perceptions of medication depend on what subjective values they attach to that concept. This is section 4 “Incorporate Interpersonal Connection Feedback”, including values such as positive affirmations and burden on loved ones.
- Knowledge management: a large-scale system that requires more complex collaborations, as presented in section 5 “Building Supportive Resources”. For reference on this aspect, the following article is highly recommended: https://www.nature.com/articles/s41599-022-01034-6
Furthermore, considering the positive impacts of medical research on not only patients but also the whole society, the authors might want to briefly mention the advocation for better attention and support from the government or other institutions toward scientific endeavors; I recommend the following article: https://www.nature.com/articles/s41562-017-0281-4
Author Response
Responses and revisions:
Thank you for the detailed information and suggestions to make our paper clearer.
- The manuscript focus on the experiences of MHNs on how to increase medication adherence in schizophrenia. There may underpin conceptual framework but nurses experience. But reviewer suggests is useful for us to application medication adherence in the future.
- Reviewer gives us a perfect suggestion to describe more clarity in the discussion section. We insert the different paragraphs made a new subsection.
- Thanks for reminding us, we added to mention the advocating for better attention and support from the government or other institutions in the last of the discussion.
Sincere thanks to the Reviewer for the insightful suggestions to rethink and adjust this research more completely. Your suggestion will help us modify the structure of the paper.
Round 2
Reviewer 4 Report
Dear authors,
Thank you very much for your revised submission. Much of the revised text is found relevant and improves the paper's readability.
Nonetheless, although you have included the relevant discussions, it appears that texts have not cited the references used, so I put them here for your proper referencing, before it can be recommended for publication.
1) Knowledge management framework: https://www.nature.com/articles/s41599-022-01034-6
2) Cost-effectiveness and risk of policy failures: https://www.nature.com/articles/s41562-017-0281-4
I look forward to your revised text again, and trust that these can be handled in no time.
Best wishes,
Author Response
Responses and revisions:
Thank you for the detailed information and suggestions to make our paper clearer.
- Reviewer gives us a perfect suggestion to increase the knowledge management framework in our study.
- 37-39 is the new reference for citation.
Sincere thanks to the Reviewer for the insightful suggestions to rethink and adjust this research more completely. Your suggestion will help us modify the structure of the paper.